# RETRACTED: A Complex Intervention Integrating Prism Adaptation and Neck Vibration for Unilateral Neglect in Patients of Chronic Stroke: A Randomised Controlled Trial

**DOI:** 10.3390/ijerph192013479

**Published:** 2022-10-18

**Authors:** Hyun-Se Choi, Bo-Min Lee

**Affiliations:** 1Department of Rehabilitation Medicine, Seoul National University Bundang Hospital, Seongnam 13620, Korea; chlgus285@naver.com; 2Department of Rehabilitation Science, Inje University Graduate School, Gimhae 50834, Korea

**Keywords:** unilateral neglect, chronic stroke, prism adaptation, neck vibration, complex intervention

## Abstract

Unilateral neglect in patients of chronic stroke reduces the quality of life and interferes with activities of daily living (ADL). This study aimed to investigate the effects of a complex rehabilitative programme that integrates prism adaptation (PA) and neck vibration (NV) for unilateral neglect in patients of chronic stroke. Thirty-six patients were randomised among the PA + NV group (Group A, *n* = 12), the NV-only group (Group B, *n* = 12), and the PA-only group (Group C, *n* = 12). The intervention was performed for 50 min/day, with five sessions per week, for 4 weeks. Albert’s test and the Catherine Bergego Scale were used to measure the effects of each intervention on unilateral neglect, whereas the modified Barthel Index was used to assess the effect on ADL. All three groups exhibited a reduction in unilateral neglect and an improvement in activities of daily living after the intervention (*p* < 0.05). Notably, Group A (PA + NV) exhibited a significantly greater level of reduction in unilateral neglect than the other groups (*p* < 0.05); however, the improvement in ADL did not significantly vary across the three groups (*p* > 0.05). This novel complex intervention comprising PA + NV is recommended for the rehabilitation, in the clinical setting, of patients of chronic stroke with unilateral neglect.

## 1. Introduction

Unilateral neglect is a consequence of stroke, wherein the patient, despite the absence of ipsilateral motor dysfunction or sensory system impairment, manifests a diminished response to stimuli on the contralateral side of the body from that of the brain injury [1]. Notably, patients with unilateral neglect experience restrictions in independently performing activities of daily living (ADL); for example, these patients may eat food only from the right side of the plate, dress themselves on only one side, and in severe cases, brush their hair only on the right-hand side [2]. Unilateral neglect is a common symptom that arises from an injury to the right hemisphere of the cerebral cortex, which is responsible for spatial concentration, and occurs in up to 85% of stroke patients with right hemispheric damage [3]. In addition to local brain lesions, unilateral neglect may occur from a dysfunction of the large-scale brain network, which is connected through white matter bundles. In the acute phase, brain damage may appear localised in areas such as the posterior parietal lobe or the middle and inferior frontal lobe [4], whereas, in the chronic phase, it may be induced by frontoparietal network dysfunction and corpus callosal disconnection [5].

Most patients with unilateral neglect achieve complete or partial recovery within 1 month of the stroke. However, despite an active rehabilitation treatment, one-third of the patients experience a persistence of symptoms into the chronic phase [6]. Rehabilitation for unilateral neglect may not achieve adequate results in patients who, owing to reduced awareness, do not actively participate in the rehabilitative treatment [7]. Furthermore, compared to other patients, those with chronic stroke and unilateral neglect may have a delayed return to daily life and a reduced quality of life [8]. With regard to the advantages and efficiency upon discharge, the Functional Independence Measure (FIM) and the modified Barthel Index (MBI) of patients of stroke with unilateral neglect are significantly low because the concurrent symptoms of unilateral neglect may impair the sense of balance and visual perception, increase the risk of falls or cause a safety problem, and confer limitations in hygiene management [9]. Therefore, rehabilitation for unilateral neglect is essential.

Luauté et al. conducted a systematic review of several existing techniques for the reduction in unilateral neglect and recommended six interventions: prism adaptation (PA), neck vibration (NV), visual scanning, video feedback training, mental imagery, and limb activation training. Among them, PA and NV are the two most widely used techniques for the rehabilitation of patients with unilateral neglect [6]. In PA, the patient wears prism goggles that shows the external environment shifted on the affected side and repeatedly points at a target. This intervention increases the spatial perception on the affected side in stroke patients with unilateral neglect [10], thereby improving the ability to perform ADL [11]. It is recommended that PA be applied as soon as possible, even in patients with chronic stroke, as PA plays a critical role in restoring the connections between the prefrontal and temporoparietal lobes [12]. In NV, vibratory stimuli are applied to the neck extensor muscles on the affected side to convert the afferent input of the peripheral sensory organs to a coordination framework of the body and core [13,14]. Furthermore, the use of only NV reduces the extent of unilateral neglect, and an additional advantage is that NV can be actively applied even without patient cooperation [15].

The two abovementioned techniques vary in their treatment mechanism. The PA-based intervention requires the active participation of the patient with a goal to improve the visuospatial representations with a focus on extra-personal factors. In contrast, the NV-based intervention does not require the active participation of the patient, and the technique aims to control or adjust neurogenesis in a framework with a displaced centre of body or core [16,17].

We hypothesised that the positive effects and treatment benefits of the two methods may be amplified in combination, as a complex intervention for treating unilateral neglect [18]. Compared to the effects of only visual scanning stimulation (VSC), the combination of VSC with transcutaneous electrical nerve stimulation (TENS) exerted greater positive effects for reducing unilateral neglect [19]. Similarly, compared to the use of either PA or functional electrical stimulation (FES), the combination of PA and FES conferred greater positive effects for symptom reduction [20]. However, these previous studies were mostly conducted on acute stroke patients, and this limitation hindered the verification of whether the improvement in unilateral neglect was owed to the treatment or whether it was attributable to spontaneous recovery.

In another previous study, wherein the use of NV alone and the combination of PA and NV were compared, the combination was shown to be more effective in the treatment of unilateral neglect [16]. However, this study had limitations due to the small sample size and the exclusion of the use of PA alone in the comparison. Additionally, the participants included both acute and chronic stroke patients.

Therefore, we aimed to determine the effects of a complex intervention to simultaneously apply PA and NV in patients with chronic stroke and unilateral neglect.

## 2. Materials and Methods

### 2.1. Study Design

This study employed a multicentre, parallel randomised controlled, single-blind design. The participants of this study were recruited among patients who were admitted to A Hospital in Seoul and to B Hospital in Gyeonggi-do, South Korea, from August 2019 to August 2020. The participants and their guardians were provided a detailed explanation of the study’s purpose, the study components/procedures, and the risk factors. The freedom to withdraw from the study at any time without any disadvantage was emphasized. All participants voluntarily provided written informed consent for study participation. In reference to the study of Kim et al. [21], the G*power programme was used to estimate a suitable sample size, effect size (d = 0.76, testing power 95%), and significance level (α = 0.05), and the participants were recruited accordingly. The sample size required in each group for the analysis of variance (ANOVA) was 12, which indicated that a total of 36 participants were required for the control and experimental groups in this study (Figure 1).

### 2.2. Participants

A total of 41 participants were recruited. After screening based on the inclusion and exclusion criteria, 36 participants were randomised into three groups.

#### 2.2.1. Inclusion Criteria

The inclusion criteria were as follows:Confirmed diagnosis of stroke based on magnetic resonance imaging (MRI) findings;Presence of stroke and absence of other diseases;A Korean version of Mini-Mental Status Examination (K-MMSE) score ≥ 20, indicating lack of cognitive, auditory, or visual impairment, and the ability to follow instructions [22];Onset of stroke at least 6 months earlier;Suspected unilateral neglect, based on the Motor-Free Visual Perception Test (MVPT).

#### 2.2.2. Exclusion Criteria

The exclusion criteria have been listed below:Participation in another rehabilitation program for unilateral neglect;Presence of a medical condition or other reasons that prevent participation in the intervention;Apraxia [23].

All tests and interventions were performed by occupational therapists who had at least 3 years of experience and were trained on the intervention methods by the principal investigator on the test.

### 2.3. Instruments

#### 2.3.1. Korean Mini-Mental State Examination

The K-MMSE is a Korean-translated version of the MMSE that comprises six categories and 27 questions. The categories include orientation, registration, attention, calculation, spatiotemporal reconstruction, and language. A total score of 30 is achievable; scores ≥ 24 indicate normal functions, whereas scores < 20 indicate a moderate level of cognitive impairment. In this study, patients with scores ≥ 20 were recruited in accordance with the inclusion criteria [22,24].

#### 2.3.2. Motor-Free Visual Perception Test

The MVPT, a standardised tool to assess the overall visuoperceptual performance, comprises 36 questions across five domains: visual discrimination, figure–ground, visual memory, visual closure, and spatial relation. The results consist of the raw score of the total number of correct responses and the response score of the number of correct responses on the left and right items. The mean time of performance for each item is measured and recorded. In this study, the response scores for the items on the left side were considered as the relevant values [25].

#### 2.3.3. Albert’s Test

Albert’s Test consists of a test paper that contains 40 randomly positioned lines. The participant watches a demonstration wherein the centre of four lines is marked and the participant is then instructed to locate the centre of the remaining lines and mark them. The test duration is unrestricted, and the test is concluded when the participant announces that all lines have been marked. In this study, the number of neglected lines among the remaining 36 lines was considered as the relevant value [26].

#### 2.3.4. Catherine Bergego Scale

The CBS assesses patients’ functions with regard to 10 ADL through monitoring and interviews, and a 4-point Likert scale is used for scoring (total score 30). Lower scores indicate a weaker impact of unilateral neglect on ADL whereas higher scores indicate a stronger impact [27,28].

#### 2.3.5. Modified Barthel Index

The MBI assesses 10 basic ADL: personal hygiene, taking a bath, use of toilet, having a meal, getting dressed, control of toileting, use of stairs, gait, use of wheelchair, and transfer. The total score is 100, and a 5-point Likert scale is used. Higher scores indicate higher levels of independence in ADL [29].

### 2.4. Experimental Procedures

Based on the inclusion criteria, 36 of a total of 41 patients were included in this study. The participants were randomly assigned to the groups A (PA + NV), B (NV), or C (PA) by using random numbers generated using a computer software (Excel, Microsoft) by an occupational therapist who was blinded to the allocation. Each participant in the three groups underwent the test on unilateral neglect before the intervention, and the test results confirmed homogeneity via statistical analysis.

Interventions were conducted 5 times per week for 50 min/day, for a total of 20 times during a 4-week rehabilitation period. For all three groups, 30 min of conventional occupational therapy were provided; in addition, for 20 min, Group A performed NV with PA on the affected neck extensor, Group B performed NV, and Group C performed PA.

The conventional occupational therapy consisted of joint exercises, task-oriented training, and ADL training. The joint exercises included passive range of motion (ROM), active-assisted ROM, and resistance exercises. The task-oriented training involved various activities, such as block building and ring assembly, which were conducted in a step-by-step manner and reflected the functional level of the patient. The ADL training involved having a meal, getting dressed, and using the toilet.

In PA, which was conducted according to the method described by Rode et al. [30], triangular prism goggles with the basal plane tilted to the left were used to ensure that the visual axis was biased toward the right side by 10°, which allowed for the wearer to perceive objects as if they had moved to the right by 10°. During PA, a blind panel was placed in front of the participant and a panel with a gradation scale was placed behind the blind panel. This was to prevent visual compensation that may arise when the patient’s arm came into the line of vision as the patient pointed at a target. The patients were guided to wear the prism goggles and then instructed to point with their contralateral hand (without movement impairment) at what they considered to be the centre of the graded panel in front of them. The patients were then guided to lower the hand and re-lift it to point at the centre again. Throughout this training, the therapist did not comment on the centre point. The patients were given a 30 s rest after every 2 min performance during the 20 min training session [31]. The treatment effect of PA is attributable to the conversion of spatial data from the retina-centred perceptual coordination toward the body-centred movement coordination for the perception of object position, which induces an adaptation effect to create a stimulation of the centre point [32].

In NV, vibratory stimuli are applied to the neck extensor muscles on the stroke-affected side using a vibration stimulator (DAS NOVAFON, Weinstadt, Germany). The device had a circular plate with a flat, smooth surface measuring 4 cm in diameter, and was set at a 0.4 mm amplitude and a 80 Hz frequency. The NV applied in this study was based on protocols from Mclntyre and Seizova-Cajic [33] and Kamada et al. [34]. The vibratory stimuli were applied to the upper sternomastoid and splenius muscles, approximately 6 cm away from the cervical vertebrae. The patient was given a 30 s rest between every 2 min stimulation.

### 2.5. Data Analysis

For data analysis, SPSS version 25.0 for Windows was used. Before conducting the intervention in each of the three groups, one-way ANOVA was performed to verify homogeneity across the groups. The Shapiro–Wilk test was performed to examine the normality of data, which was confirmed before parametric analyses were applied. The participants’ general characteristics were analysed through frequency analysis, descriptive statistics, and ANOVA.

The paired *t*-test was performed to ascertain the effects from before and after the intervention within each group, and one-way ANOVA was used to compare the intergroup differences in the effects of the intervention. For the post hoc analysis, Scheffe’s test was performed. The level of significance (α) was set at *p* = 0.05.

## 3. Results

### 3.1. General Characteristics of the Participants

The general characteristics of the participants in this study are presented in Table 1. The study cohort comprised 36 participants, with 12 participants in each group. No significant differences among the general characteristics were noted across the three groups.

### 3.2. Differences from before to after the Intervention

After the intervention, the participants in all the three groups exhibited a significant improvement in the scores of the MBI, Albert’s test, and the CBS. On the intergroup comparison, Group A exhibited a trend of a higher increase in MBI, although this was not significant (*p* < 0.05). For the scores of Albert’s test and the CBS, significant intergroup differences were found (*p* < 0.01 and *p* < 0.05, respectively); there were non-significant differences in the comparison between groups B and C and there were significant differences with Group A (see Table 2).

## 4. Discussion

This study determined the effects of a complex rehabilitative intervention that integrates NV and PA for the treatment of unilateral neglect in patients of chronic stroke. When the three groups were compared before and after the intervention, a reduction in unilateral neglect and an improvement in ADL were found. Our findings are similar to those of studies by Mizuno et al. [35] and Kamada et al. [34], wherein the results suggested that the combination of PA and NV reduces unilateral neglect and improves ADL.

According to Jeong and Park [36], for PA, the angle of the prism goggles is to be tilted at 10°; however, Chen et al. [37] suggested that PA may reduce unilateral neglect when the treatment period and intensity were increased. Similarly, in our study, 10° tilted prism goggles were used, and as PA was performed every day for 4 weeks, unilateral neglect in patients was reduced, as predicted.

In NV, the treatment mechanism is not as simple as the provision of external stimulation on the stroke-affected side to improve attention, and instead targets gradual improvements in spatial representation on the neglected side that leads to treatment effects [38]. The neck muscles, compared to the limb muscles, show a remarkably high spindle index, with a high diversity of fibre types that relay positional information regarding posture. Thus, it is appropriate to target the neck muscles in an intervention for unilateral neglect [34,39]. We found that NV elicited a compensatory reaction to adjust the central line of the body that had been tilted towards the right side alongside the sensory impairment due to the neurological damage to the midline, and this improved the ability to explore the space on the left side, thereby ensuring the reduction in unilateral neglect.

The intergroup comparison of the intervention effects showed that the level of reduced unilateral neglect was higher in Group A than in groups B and C, whereas no significant difference was observed in terms of improvement in ADL. A previous study of 12 stroke patients with unilateral neglect found that a combination therapy with 20 min of PA and NV produced more synergistic intervention effects compared to that of NV alone [16]. In another study conducted on 30 patients of acute stroke with unilateral neglect, the combined effects of PA and FES were greater than the effects of either intervention alone [20]. In a study that applied a complex intervention, which combined an eye patch and optokinetic stimulation, positive effects were shown in terms of the reduction in unilateral neglect and improved ADL, compared to the control [40]. The results of our study coincide with those of previous studies in terms of reduction in unilateral neglect but not in terms of improvements in ADL. It is presumed that the intergroup differences of the CBS and MBI were not significant because the CBS aims to monitor activities that are directly influenced by unilateral neglect, such as personal hygiene, having a meal, getting dressed, and using a wheelchair, whereas the MBI mainly monitors activities that are indirectly influenced by unilateral neglect, such as control of toileting, using stairs, and transfer.

Unilateral neglect is caused by complex neuropathophysiological mechanisms. Considering the diversity of problems in the sensory system and the spatial representation and concentration, it may be necessary to adopt an approach that is based on a complex intervention [41]. For a complex intervention, it is recommended that two methods of different mechanisms be combined [18], especially when the methods can reinforce visual scanning and kinesthesis [41]. Unlike methods that rely on prosthesis or therapist assistance, prism adaptation is a corrective active treatment approach, wherein the patients exert concentration and independent participation as they point at the target [2]. In addition, PA is an intervention that targets the neglect of extra-personal factors, with positive effects noted on spatial unilateral neglect involving spatial scanning and mental representation [42]. NV is a passive treatment method that necessitates assistance from a therapist, wherein the neck muscles are stimulated to activate the motor senses and re-form the central line as an intervention for unilateral neglect [16,43]. Accordingly, the brain areas that are activated in PA and NV vary. In PA, the activation is mainly shown by the left temporal medial cortex, the left temporooccipital cortex, the right posterior parietal cortex, the right cerebellum, and the frontoparietal connection [44]. In NV, the activation is shown by the somatosensory area of the right insula and right lateral sulcus [45]. Furthermore, the PET result shows the activation of numerous regions of the damaged brain, following the intervention that combined PA and NV [16]. Likewise, the complex intervention that combined two techniques with different mechanisms of action in this study showed synergistic therapeutic effects rather than the sum of the effects of either technique alone.

Compared to previous studies, this study is significant in three aspects: first, a suitable sample size for determining the interventional effects was satisfied; second, a complex intervention without any influence of spontaneous recovery was applied to patients of chronic stroke; and third, a randomised controlled trial design provides a higher level of evidence than single-centre or case-series studies do.

This study verified the positive effects of the complex intervention of PA and NV, which constitute two widely applied methods in clinical settings, on the reduction in unilateral neglect in patients of chronic stroke. The significance of the study is that it provides basic data to support the combined application of PA and NV in clinical practice.

This study had some limitations. First, the continuity of the interventional effects could not be verified owing to the lack of follow-up monitoring. Second, the effects of conventional occupational therapy on unilateral neglect cannot be ruled out. Third, the activation of the brain hemisphere was not verified using a neuroimaging device to confirm the effects on unilateral neglect. Further studies should thus be conducted to resolve these limitations. In addition, the effects of combining different rehabilitation techniques other than PA and NV on unilateral neglect should be determined in chronic stroke patients via comparative analyses.

## 5. Conclusions

This study verified the effects of a complex programme of PA and NV on unilateral neglect in patients of chronic stroke. The comparison from before to after the intervention showed positive effects in all three groups with regard to the reduction in unilateral neglect and the improvement in ADL, and demonstrated more synergistic effects with the complex intervention of PA and NV than with either technique alone. The results verified the potential use of the complex intervention in patients of chronic stroke in the clinical setting.

## Figures and Tables

**Figure 1 ijerph-19-13479-f001:** Flow chart of the participant selection process in this study.

**Table 1 ijerph-19-13479-t001:** General characteristics of the participants before the intervention.

Characteristics	Group A	Group B	Group C	*p*
Sex	Male	7	4	6	0.886
Female	5	8	6
Onset (months)	8.92 ± 2.64	9.70 ± 3.16	9.37 ± 2.85	0.78
Age (years)	62.90 ± 8.64	67.70 ± 9.76	66.00 ± 12.09	0.58
K-MMSE (score)	22.40 ± 1.47	23.60 ± 1.26	22.03 ± 1.71	0.26
MVPT (score)	6.91 ± 1.66	6.63 ± 2 = 1.38	7.21 ± 2.30	0.82
Cause of injury	Cerebral haemorrhage	8	5	9	0.87
Cerebral infarction	4	7	3

Data in the table are the mean ± standard deviation or count, unless specified otherwise. Group A: PA + NV, Group B: NV, Group C: PA. K-MMSE: Korean version of Mini-Mental Status Examination; MVPT: Motor-Free Visual Perception Test.

**Table 2 ijerph-19-13479-t002:** Intergroup comparison from before to after the intervention.

Variables	Group A (a)(*n* = 12)	Group B (b)(*n* = 12)	Group C (c)(*n* = 12)	F	*p*
MBI					0.333
Pre-test	47.69 ± 2.75	49.30 ± 2.49	46.30 ± 4.29	1.145
Post-test	58.10 ± 2.76	58.30 ± 3.65	55.60 ± 3.50	
Within-group changest	10.50 ± 2.71(−12.215) **	9.10 ± 2.53(−11.211) **	9.30 ± 1.63(−17.972) **	
Albert’s test				9.398(a > b, c)	0.001 **
Pre-test	7.80 ± 1.75	7.30 ± 0.95	7.80 ± 1.75
Post-test	5.00 ± 2.21	5.70 ± 1.48	5.00 ± 2.21
Within-group changest	2.80 ± 0.91(10.811) **	1.60 ± 0.84(6.002) **	1.30 ± 0.67(6.091) **
CBS					0.041 *
Pre-test	20.90 ± 2.99	19.00 ± 2.98	20.10 ± 2.77	3.598(a > b, c)
Post-test	14.70 ± 1.70	14.80 ± 3.10	15.40 ± 2.13	
Within-group changest	6.20 ± 1.85(9.635) **	4.20 ± 1.81(7.324) **	4.70 ± 1.89(9.485) **	

Group A: PA + NV, Group B: NV, Group C: PA. MBI: modified Barthel Index; CBS: Catherine Bergego Scale. Values are expressed as mean ± SD * Significant difference (*p* < 0.05). ** Significant difference (*p* < 0.01).

## Data Availability

The datasets used and analysed during the current study are available from the corresponding author upon reasonable request.

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
