# Peer review of "A Complex Intervention Integrating Prism Adaptation and Neck Vibration for Unilateral Neglect in Patients of Chronic Stroke: A Randomised Controlled Trial"

_ijerph, 2022, doi:10.3390/ijerph192013479_

Round 1
Reviewer 1 Report
This article concerns prism adaptation and neck vibration for unilateral neglect in patients with chronic stroke.
The topic is exciting and of scientific relevance, however, it cannot be accepted as it is. Extensive English language editing is necessary, and authors should rewrite much of the introduction as there are several inaccuracies.
From the initial sentence in which neglect is described as a "symptom" of stroke, a concept that in the medical field has a precise meaning and is not relevant in this context.
The methods should also be better explained and the procedure adopted from the first reading made understandable.
Indeed, the effect size of the sample size should not be estimated on the basis of a previous study by the first author (in which perhaps the same sample size was used), furthermore, the complete results of this analysis should be reported.
The randomization of the participants should be explained with an appropriate scientific method that is not simply "random".
The experimental procedure should be simplified and shortened, if necessary a table could be inserted to organize the data in a more schematic way.
The size of the effect is missing from the results of the statistics, without which it is not possible to establish the size of each factor being analyzed.
Author Response
Dear reviewer1,
First, thank you for your good advice.
I have revised the advice you gave me, so please review it.
Have a nice day.
Kind regards,

Reviewer 2 Report
Dear Authors,
I appreciate your effort to prepare your manuscript in its current form. The manuscript contains an interesting research idea and an indication of an important method (vibration) that can be used to work with unilateral neglect in patients with chronic stroke. I would suggest a somewhat broader description of the role of local vibration in the treatment of pain. For this purpose, you can use, for example:
1) Musumeci, G. The Use of Vibration as Physical Exercise and Therapy. J. Funct. Morphol. Kinesiol. 2017, 2, 17
2) Choi, W.; Han, D.; Kim, J.; Lee, S. Whole-Body Vibration Combined with Treadmill Training Improves Walking Performance in Post-Stroke Patients: A Randomized Controlled Trial. Med. Sci. Monit. 2017, 23, 4918–4925.
3) Zurek G., Kasper-Jędrzejewska M., Dobrowolska I., Mroczek A., Delaunay G., Ptaszkowski K., Halski T.: Vibrating Exercise Equipment in Older Women With Chronic Low Back Pain and Effects on Pain Intensity, Range of Motion and Bioelectrical Activity: A Randomized, Single-blinded Sham Intervention Study. Biology 2022, 11, 268.
4) Celletti, C.; Suppa, A.; Bianchini, E.; Lakin, S.; Toscano, M.; La Torre, G.; Di Piero, V.; Camerota, F. Promoting post-stroke recovery through focal or whole body vibration: Criticisms and prospects from a narrative review. Neurol. Sci. 2020, 41, 11–24.
I believe that the inclusion of additional publications will enrich the value of the work.
Author Response
Dear reviewer2,
First, thank you for your good advice.
I have revised the advice you gave me, so please review it.
Have a nice day.
Kind regards,

Reviewer 3 Report
In this work, they compared the effect of prism adaptation and neck vibration with respect to the two techniques alone on chronic stroke survivors with unilateral neglect. They used a combination of three tests/scales to investigate the effect of the intervention. They found that all three groups improved on the daily living activities scale and the group with the combination of treatments showed a greater level of reduction in unilateral neglect than the others.
The study is interesting and well written, but I have some minor doubts.
PA has been already used with FES in literature, showing promising results. Can you elaborate more on the advantages of using the combination of PA and NV with respect to the use of one of them in combination with other techniques like FES, VSC, and TENS.
Is there a specific reason why you define the combination of PA and NV as 'complex intervention'?
When was performed the clinical assessment with respect to the experimental procedure? Did you repeat the clinical assessment also after the end of the experimental procedure?
Was the evaluation (Albert's test, MBI, CBS) performed immediately before the start/after the end of the first and last experimental session?
Was this evaluation performed also in between the 5 sessions?
How did you choose the frequency and the number of treatments?
It is not clear to be how the between-group statistical analysis in 3.1. is done. Did you run the ANOVA on the differences between the before and after? Did you check if the before was different between the groups?
Since the two treatments alone are very different, in your opinion, why did they lead to the same improvements in all the metrics?
I think that the title of sec. 2.2.2 should be 'Exclusion criteria'.
In the manuscript, some abbreviations are reported multiple times like K-MMSE, and MVPT, while others are defined and never used. In addition, in the footnotes of Table 1 you reported 'NV: neck vibration; PA: prism adaptation.', but not used in the table.
Author Response
Dear reviewer3,
First, thank you for your good advice.
I have revised the advice you gave me, so please review it.
Have a nice day.
Kind regards,
